# Magnitude and antimicrobial susceptibility profiles of Gram-Negative bacterial isolates among patients suspected of urinary tract infections in Arba Minch General Hospital, southern Ethiopia

**Asaye Mitiku, Addis Aklilu\*, Tsegaye Tsalla, Melat Woldemariam, Aseer Manilal[ORCID]\*, Melkam Biru**

Department of Medical Laboratory Science, College of Medicine and Health Sciences, Arba Minch University, Arba Minch, Ethiopia

\* aseermanilal@gmail.com (AM); addaklilu@gmail.com (AA)

## Abstract

The emergence of drug-resistant Gram-negative bacterial uropathogens poses a grave threat worldwide, howbeit studies on their magnitude are limited in most African countries, including Ethiopia. Therefore, measuring the extent of their drug resistance is essential for developing strategies to confine the spread. A cross-sectional study was conducted at title hospital from 01 June to 31 August 2020. Midstream urine specimens were collected and inoculated onto MacConkey agar. Positive urine cultures showing significant bacteriuria as per the Kass count ($>10^5$ CFU/mL) were further subjected to biochemical tests to identify the type of uropathogens. Antimicrobial susceptibility testing was performed by the Kirby-Bauer disk diffusion technique, and potential carbapenemase producers were phenotypically determined by the modified carbapenem inactivation method as per the CLSI guidelines. Data were analyzed using SPSS version 26; P-value <0.05 was considered statistically significant. Totally, 422 patients were included, and the majority were females (54.7%). The prevalence of carbapenem-resistant Gram-negative uropathogens was 12.9%, and 64.7% of them were carbapenemase producers. *Klebsiella pneumoniae* (n = 5) was the predominant carbapenemase producer, followed by *Pseudomonas aeruginosa* (n = 4). Consumption of antibiotics prior to six months of commencement of the study, the presence of chronic diseases and hospitalizations were statistically associated with UTI caused by carbapenem-resistant Gram-negative uropathogens. Carbapenemase producers were resistant to most of the antibiotics tested. Our findings highlight the need for periodic regional bacteriological surveillance programs to guide empirical antibiotic therapy of UTI.

**Data Availability Statement:** All relevant data are within the paper and its Supporting Information files.

**Funding:** The author(s) received no specific funding for this work.

**Competing interests:** The authors have declared that no competing interests exist.

# Introduction

Urinary tract infections (UTIs) are the second most common bacterial infection affecting all ages, irrespective of sex, in the community, whereas they are the most common hospital-acquired infections [1]. It is labelled as the second most common cause of bacteremia in hospitalized patients [1]. Nowadays, UTI is considered a serious global public health menace, and the annual incidence amounts to almost 150 million, resulting in healthcare costs of 6 billion US $ [2]. Gram-negative bacteria have been implicated as the prominent and putative causative agents of UTI, including the members of Enterobacteriaceae such as *Escherichia coli*, *Klebsiella* spp., *Enterobacter* spp., *Citrobacter* spp. and non-fermenting organisms, particularly *Pseudomonas aeruginosa* [3].

The majority of un-complicated UTI cases are treated empirically with broad-spectrum antibiotics, and 15% of all antibiotics prescribed in the community are used to treat UTIs, and this corresponds to over usage [4]. In recent decades, an upsurge in the evolution and spread of multidrug-resistant Gram-negative bacteria, such as extended-spectrum beta-lactamases (ESBL) and carbapenem-resistant Enterobacteriaceae (CRE), have been reported worldwide [5]. The emergence of carbapenem resistance among Gram-negative bacteria is a significant concern since carbapenem currently is the treatment of choice for severe infections caused by multidrug-resistant (MDR) strains producing ESBLs [6]. The Centers for Disease Control and Prevention (CDC) defines CRE as any member of the Enterobacteriaceae family resistant to imipenem, meropenem, doripenem, or ertapenem [7]. Most carbapenemases are plasmid-mediated and have been found primarily in Enterobacteriaceae [8]. A thorough literature survey indicates that UTIs are among the top listed infections caused by carbapenem-resistant Gram-negative bacteria (CRGNB) [9, 10]. In addition, high mortality rates (ranging from 30 to 75%) are characteristics of severe CRE infections [11]. Currently, CRGNB are tough to manage and pose a grave threat to the healthcare system as there are only limited therapeutic options [12]. Consequently, CRE and carbapenem-resistant *P. aeruginosa* are enlisted as the priority one critical pathogen by WHO [13]. Besides, the emergence and the gradual augmentation of carbapenem resistance can be correlated to several associated factors like exposure to UTI, history of usage of antibiotics, hospitalization, and the presence of chronic underlying diseases [14].

The burden of infections caused due to CRGNB is massive in African countries, including Ethiopia [15]. However, in these countries, there exist a severe shortage of trained laboratory personnels. Also, inadequate infrastructure in microbiology laboratories, paucity of consumables required for diagnosis, and the under-appreciation of the relevance of laboratory data by the prescriber to arrive at the correct antimicrobial choice are the main factors that impede the appropriate treatment of infections [15]. Apart from the empirical use of antimicrobials in clinical settings in Ethiopia, the absence of strict regulations in sales is also a driving factor related to the access and misuse of antimicrobials in both urban and rural parts of the country [16]. Another vexing problem is that local antimicrobial resistance recommendations for the empirical prescription of antibacterial for UTIs were not considered at all in formulating the standard treatment guidelines [17]. This leads to poor diagnosis and irrational usage of antibiotics causing the emergence and worsening the spread of multidrug-resistant bacteria.

Recently, the Infectious Diseases Society of America recommended that empirical antibiotic treatment for UTIs should be based on regional susceptibility data, drug accessibility, and patient history [18]. Therefore, knowledge of the local epidemiology of multidrug-resistant bacteria is a key tenet in determining the empirical antimicrobial therapy and assisting in escalation and de-escalation wherever possible. However, a literature survey indicated a sheer shortage in the quantum of work conducted on CRGNB among patients suspected of UTI,

particularly in Ethiopia [19]. Hitherto, no research has been done in this context in the study area, Arba Minch. Moreover, the number of cases of uncomplicated UTI is sharply rising in the title hospital, and due to the lack of prompt culture facilities, empirical treatments are given exclusively based on clinical criteria. Hence we anticipate a higher magnitude of drug-resistant bacterial uropathogens, including the CRGNB. Therefore, this study aims to determine the magnitude and antibiogram of Gram-negative uropathogens with particular emphasis on CRGNB among patients suspected of urinary tract infections attending Arba Minch General Hospital, Arba Minch, southern Ethiopia. This baseline knowledge may provide vital information required to update the antibiotic policy for managing UTIs in the title hospital.

## Materials and methods

### Study area, design period, and population

An institution-based cross-sectional study was conducted from 01 June to 31 August 2020 in AMGH, located in Arba Minch, Gamo Gofa Zone. This hospital was established in 1961 and serves more than 1.5 million by providing preventive and curative care in outpatient and inpatient departments (OPD and IPD). According to the data obtained from the Health Management Information System, the average annual flow of patients corresponds to 90,000 in the OPD, 12,000 in the IPD, 2,000 in the surgical ward, 1800 in the medical ward, 7,000 in the emergency OPD, and 6,800 in the minor OPD. The study population comprises adult patients suspected of UTI and who attended the OPD of AMGH during the above-mentioned period. Therefore, the inclusion criterion is all adult patients clinically suspected of uncomplicated UTI (i.e., the presence of at least two of the following clinical symptoms- frequency, urgency, dysuria and hematuria) who presented at the OPD of AMGH during the data collection period. The exclusion criteria involve the following: 1. patients who had received antibiotics within the previous two weeks of the commencement of the study, 2. juveniles (whose age was less than 18), 3. those who were critically ill. The study was ethically approved by the Institutional Review Board of the College of Medicine and Health Sciences, Arba Minch University (Decision number of ethics committee approval IRB/183/12/03/2020).

### Sample size determination and sampling technique

The sample size was computed using a formula to estimate a single population proportion. A proportion of 50% of the population was selected to determine the sample size due to the paucity of data derived from previous studies. After considering a 95% of confidence interval (z = 1.96) and 5% of marginal error (d = 0.05), the initial sample size estimated was 384, and by computing a 10% of non-response rate (38 subjects), the final sample size was consolidated as 422. A systematic random sampling technique has opted; considering the number of patients corresponding to the previous year (~570), the $K^{th}$ value was calculated, and individuals were selected by a lottery method.

### Collection of demographic and clinical data

Written informed consent was obtained from all the subjects who participated in the study. A structured questionnaire was used as the data collection tool. The questionnaire was translated into Amharic (the native language) and was administered through a face-to-face interview to collect the socio-demographic characteristics. In addition, clinical data such as the status of any chronic underlying diseases (diabetes mellitus, hypertension and renal problems), previous episodes of UTI, and duration of hospitalization were reviewed from the medical records of patients (supplemented by interviews with patients). Medical records of the patients were

reviewed during the data collection, and their card number was used as unique codes to distinguish the entries.

## Sample collection, isolation, and identification

Mid-stream urine samples (10–20 ml) were collected with all aseptic precautions as described elsewhere [20]. Collected samples were labelled and immediately transported at ambient temperature to the Medical Microbiology and Parasitology Laboratory, Department of Medical Laboratory Science, College of Medicine and Health Sciences, Arba Minch University. Urine specimens were inoculated using a disposable calibrated plastic loop (1µL) onto MacConkey agar. All the streaked culture plates were then incubated at 37˚C for 24 to 48 hrs. The samples displaying prominent bacterial growths according to the Kass count ($>10^5$ CFU/mL) were considered culture-positive for UTI [21]. Isolation and identification were done according to the standard bacteriological protocols; colony morphology, Gram reaction, and microscopic features were used as the primary identification criteria. The identity of bacterial isolates was further confirmed by an array of conventional biochemical analyses [22].

## Antimicrobial susceptibility testing

The antimicrobial susceptibilities of bacterial isolates were tested according to the criteria set by the Clinical and Laboratory Standards Institute using the Kirby–Bauer disk diffusion method on Mueller-Hinton agar (Oxoid, UK) [23]. Nine commercially available antibiotic disks (Himedia, India) were used, such as carbapenems (meropenem (10µg) and imipenem (10µg)), fluoroquinolone (ciprofloxacin (5µg) and norfloxacin (10 µg)), amino-penicillin (ampicillin (10µg)), anti-pseudomonal penicillin (piperacillin (100 µg)), aminoglycosides (tobramycin (10µg)), nitrofuran (nitrofurantoin (300 µg)), antifolate (trimethoprim-sulfamethoxazole (1.25/23.75 µg)). These were selected based on CLSI guidelines and also by considering the national antibiotic policy. The results were interpreted according to the CLSI (2020) guidelines [24]. For a convenient statistical analysis, intermediate and resistant isolates put together were considered as a single entity. Further analysis was conducted to calculate the multidrug resistance [25] and was assessed in relation to the AWaRe classification [26].

## Detection of carbapenem-resistant and carbapenemase-producing uropathogens

Carbapenem (imipenem or meropenem) intermediate or resistant organisms were further subjected to a modified carbapenem inactivation method (mCIM), for the confirmation of carbapenemase enzyme production [27].

## Data quality control

To ensure the quality of data, 5% of samples (n = 21) were pre-tested before the actual work. Quality control measures were maintained throughout the process of data and specimen collection in order to ensure the reliability of results. Staining reagents, culture media, and antibiotic disks were inspected for their normal shelf life. All culture plates and antibiotics were stored at the recommended refrigeration temperature (2–8˚C). Reference strains of *S. aureus* (ATCC 25923), *E. coli* (ATCC 25922), and *P. aeruginosa* (ATCC 27853) were used as controls which were obtained from Ethiopian Public Health Institute.

## Statistical analysis

The data were analyzed using SPSS, Chicago, IL, the USA for Windows, version 26. Descriptive statistics, including frequency, mean and standard deviations, were performed. Bivariable and multivariable logistic regression analyses were performed to evaluate the association among variables and the CRGNB uropathogens. Variables with a *P*-value <0.25 in the bivariable logistic regression model were subsequently analyzed in the multivariable logistic regression to control the confounding factors, and a *P*-value ≤0.05 from multivariable logistic regression was considered statistically significant.

## Results

### Socio-demographic characteristics and clinical profiles

A total of 422 UTI-suspected participants were enrolled in this study. Of these, 54.7% (231/422) were females. The mean age of study participants was 39.9±16.8, and the range of age was 18–90. Approximately one-fourth of them were in the age group of 18–26, i.e., 25.8% (109/422). Urban residents and students who completed secondary school constituted 67.5% (285/422) and 42.9% (181/422), respectively. Among the participants, 29.9% (126/422) were government employees, and they earned a monthly income of more than 2001 Ethiopian birr (Table 1). Among a total of 422 study participants, 62.6% (264/422) had a history of receiving antibiotics during the previous six months of the commencement of the study, while more than half of them, 54.5% (230/422), had a history of UTI. More than one-fourth of the participants, i.e., 29.6% (125/422) had chronic underlying diseases (the frequently associated conditions included diabetes mellitus (49.6%) followed by hypertension (30.4%) and renal problem (20%)), and 36.0% (152/422) had an experience of hospitalization during twelve months prior to the initiation of the study (Table 2).

### Bacteriological profile

Among the 422 urine samples, only 129 cultures were found to have significant bacteriuria, and a total of 131 Gram-negative bacteria were isolated, making an overall prevalence of 30.6% [95% CI:(33.7–38.3)]. Two urine cultures showed mixed growths of bacteria (*E. coli* with *Proteus mirabilis* and *E. coli* with *K. pneumoniae*). Of all the isolated organisms, *E. coli* corresponded to the highest frequency, 32.1% (42/131), followed by *Klebsiella pneumoniae* at 29.8% (39/131), whereas *Citrobacter* sp. 3.1% (4/131) were the least isolated. In this study, the prevalence of UTIs caused by Gram-negative bacteria was 30.6% (129/422); UTIs were slightly more frequent among females, 35.1% (81/231) compared to males, 26.2% (50/191), and also the frequency was found to be higher among the younger age group, i.e., 16.5%(38/231) (Table 3).

### Antimicrobial susceptibility profiles

All the isolated uropathogens were tested for susceptibility against nine antibiotics. Gram-negative uropathogenic isolates showed varied susceptibility profiles to the commonly prescribed antibiotics for treating UTIs. The highest resistance was observed in the case of ampicillin (94.7%), followed by tobramycin (65.7%) and piperacillin (64.9%). With regard to the susceptibility profiles, 96.2 and 90.8% of isolates were susceptible to imipenem and meropenem, respectively. Among the common antibiotics used for treating UTIs, nitrofurantoin was the most effective, and 78.6% of isolates were susceptible to it. The second most effective drug was trimethoprim-sulfamethoxazole, against which 70.9% of the isolates were susceptible.

Among the total number of isolates of *E. coli*, 100% were resistant to ampicillin, whereas the extent of resistance to ciprofloxacin, trimethoprim-sulfamethoxazole, and nitrofurantoin,

**Table 1. Socio-demographic characteristics of study participants attending AMGH, southern Ethiopia, 2020.**

| Variable | Categories | Number (n) | Percentage (%) |
|---|---|---|---|
| Sex | Male | 191 | 45.3 |
| | Female | 231 | 54.7 |
| Age group | 18–26 | 109 | 25.8 |
| | 27–35 | 96 | 22.7 |
| | 36–44 | 63 | 14.9 |
| | 45–53 | 67 | 15.9 |
| | >54 | 87 | 20.6 |
| Residence | Urban | 286 | 67.8 |
| | Rural | 136 | 32.2 |
| Educational status | Unable to read and write | 37 | 8.7 |
| | Primary | 115 | 27.3 |
| | Secondary | 181 | 42.9 |
| | College and university | 89 | 21.1 |
| Marital status | Single | 120 | 28.4 |
| | Married | 226 | 53.6 |
| | Divorced | 50 | 11.8 |
| | Widowed | 26 | 6.2 |
| | Government employee | 126 | 29.9 |
| Occupation | Merchant | 93 | 22.0 |
| | Farmer | 38 | 9.0 |
| | Student | 90 | 21.4 |
| | Home maker | 47 | 11.1 |
| | Others | 28 | 6.6 |
| Monthly income Ethiopian birr | ≤1000 | 29 | 6.9 |
| | 1001–2000 | 54 | 12.8 |
| | > 2001 | 151 | 35.8 |

were 38.1, 33.3, and 26.2% respectively. At the same time, 97.6 and 92.3% of isolates were susceptible to imipenem and meropenem, respectively. The second most prevalent bacterial isolate, *K. pneumoniae*, showed 84.6, 79.5 and 59% resistance to piperacillin, tobramycin, and ciprofloxacin, respectively. Also 92.3 and 87.2% of isolates were susceptible to imipenem and meropenem, respectively. The third most dominant isolated organism, *P. mirabilis*, was 100% susceptible to both imipenem and meropenem. On the other hand, 95.2 and 76.2% of isolates were resistant to ampicillin and piperacillin, respectively. Isolates of *P. aeruginosa* showed varying degrees of resistance to drugs such as ampicillin (100%), piperacillin (87.5%), and

**Table 2. Frequency of clinical profiles of study participants attending AMGH, southern Ethiopia, 2020 (n = 422).**

| Variables | Categories | Frequencies | Percentage (%) |
|---|---|---|---|
| Previous history of UTI | Yes | 230 | 54.5 |
| | No | 192 | 45.5 |
| History of antibiotic usage in the previous six months | Yes | 264 | 62.6 |
| | No | 158 | 37.4 |
| Presence of chronic underlying diseases | Yes | 125 | 29.6 |
| | No | 297 | 70.4 |
| Hospitalization within the last twelve months | Yes | 152 | 36.0 |
| | No | 270 | 64.0 |

**Table 3. Frequency and type of GN uropathogens isolated from the urine specimen of study participants attending AMGH, southern Ethiopia, 2020 (n = 131).**

| S.N | Isolated bacteria | N | Percent (%) |
|---|---|---|---|
| 1 | E. coli | 42 | 32.1 |
| 2 | K. pneumoniae | 39 | 29.8 |
| 3 | P. mirabilis | 21 | 16.0 |
| 4 | P. aeruginosa | 16 | 12.2 |
| 5 | Enterobacter sp. | 9 | 6.8 |
| 6 | Citrobacter sp. | 4 | 3.1 |
| | Total | 131 | 100 |

tobramycin (81.2%). The percentage of susceptibility was the highest towards imipenem (93.8%), followed by meropenem (75%). All the isolates of *Enterobacter* sp. and *Citrobacter* sp. were susceptible to imipenem, meropenem, nitrofurantoin, and norfloxacin. At the same time, 55.6% of isolates of *Enterobacter* sp. were resistant to ampicillin, whereas 50% of isolates of *Citrobacter* sp. were resistant to the same drug (Table 4).

## Multi-drug resistance profiles

In this study, MDR is inferred as the resistance to three or more groups of antibiotics tested. Out of the 131 total bacterial isolates, 66 (i.e., 50.38%) were MDR. The MDR bacteria comprise 21/42 (50%) of *E. coli*, 10/21(47.6%) of *P. mirabilis*,1/9(11.11%) of *Enterobacter* spp., 2/4(50%) of *Citrobacter* sp., 22./39 (56.4.%) of *K. pneumoniae* and 10/16 (62.5%) of *P. aeruginosa*.

**Table 4. Antimicrobial susceptibility profiles of GN uropathogens isolated from patients suspected of UTI attending AMGH, southern Ethiopia, 2020 (n = 131).**

| Antimicrobial | Patterns | E. coli n = 42 (%) | K. pneumoniae n = 39 (%) | P. aeruginosa n = 16 (%) | P. mirabilis n = 21 (%) | Enterobacter sp. n = 9 (%) | Citrobacter sp. n = 4 (%) | Total |
|---|---|---|---|---|---|---|---|---|
| IMP[w] | S | 41(97.6) | 36(92.3) | 15(93.8) | 21(100) | 9(100) | 4(100) | 126(93.2) |
| | R | 1(2.4) | 3 (7.7) | 1(6.2) | 0 | 0 | 0 | 5(3.8) |
| MER[w] | S | 39(92.3) | 34(87.2) | 12(75) | 21(100) | 9(100) | 4(100) | 119(90.8) |
| | R | 3(7.7) | 5(12.8) | 4(25) | 0 | 0 | 0 | 12(9.2) |
| NIT[A] | S | 31(73.8) | 28(71.8) | 11(67.8) | 20(95.2) | 9(100) | 4(100) | 103(78.6) |
| | R | 11(26.2) | 11(28.2) | 5(32.2) | 1(4.8) | 0 | 0 | 28(21.4) |
| SXT[A] | S | 28(66.7) | 25(64.1) | 10(62.5) | 17(80.1) | 9(100) | 3(75) | 92(70.9) |
| | R | 14(33.3) | 14(35.9) | 6(37.5) | 4(19.9) | 0 | 1(25) | 39(29.1) |
| NOR[w] | S | 24(57.1) | 27(69.2) | 9(56.3) | 13(61.9) | 9(100) | 4(100) | 86(65.6) |
| | R | 18(42.9) | 12(30.8) | 7(43.7) | 8(38.1) | 0 | 0 | 45(34.4) |
| CPR[w] | S | 26(61.9) | 16(41.0) | 8(50) | 11(52.4) | 8(88.9) | 1(25) | 70(53.4) |
| | R | 16(38.1) | 23(59.0) | 8(50) | 10(47.6) | 1(11.1) | 3(75) | 61(46.6) |
| TOB[w] | S | 29(69.0) | 8(20.5) | 3(18.8) | 9(42.9) | 7(77.8) | 2(50) | 58(44.3) |
| | R | 13(31.0) | 31(79.5) | 13(81.2) | 12(57.1) | 2(12.2) | 2(50) | 73(65.7) |
| PIP[w] | S | 27(64.3) | 6(15.4) | 2(12.5) | 5(23.8) | 3(33.3) | 3(75) | 46(35.1) |
| | R | 15(35.7) | 33(84.6) | 14(87.5) | 16(76.2) | 6(66.7) | 1(25) | 85(64.9) |
| AMP[A] | S | 0 | NT | 0 | 1(4.8) | 4(44.4) | 2(50) | 7(5.3) |
| | R | 42(100) | NT | 16(100) | 20(95.2) | 5(55.6) | 2(50) | 124(94.7) |

Note: S: susceptible; R: resistant; IMP: Imipenem; MER: Meropenem; NIT: Nitrofurantoin; SXT: Trimethoprim-sulfamethoxazole; NOR: Norfloxacin; CPR: Ciprofloxacin; TOB: Tobramycin; PIP: Piperacillin; AMP: Ampicillin; NT: Not done; A: Aware group; W: watch group of antibiotics.

According to the WHO AWaRe categories, the range of resistance was as follows: Access, 21–94%; Watch, 9–65%, and Reserve, 0% (Table 5).

## Prevalence of carbapenemase producers

Of the 131 Gram-negative uropathogenic isolates, 14% (17) were resistant to either imipenem or meropenem. Of them, 29.4% (5/17) were imipenem resistant, whereas the remaining 70.6% (12/17) were meropenem resistant. Those isolates that were imipenem and meropenem resistant were considered carbapenem-resistant and therefore were subjected to further carbapenemase production tests. About 64.7% (11/17) of them were found to be carbapenemase producers. The overall prevalence of carbapenem-resistant Gram-negative uropathogens was 12.98% (17/131). The ratio of carbapenemase producers to non-producers was 1.83:1(11/6) (Table 5).

Among different species of Gram-negative uropathogens, the distribution of carbapenem resistance varied considerably. The highest frequency of carbapenem resistance was observed in the case of *K. pneumoniae*, 47.1% (8/17) followed by *P. aeruginosa*, 29.4% (5/17). The maximum number of bacterial isolates and a higher proportion of carbapenem-resistant Gram-negative isolates were retrieved from females 58.8% (10/17), and only 35.3% (6/17) were obtained from married participants. The spectrum of carbapenem-resistant Gram-negative uropathogens varied with the age of patients. The highest percentage, i.e., 35.3% (6/17) of isolates, was found among the 18–26. Of the total carbapenem-resistant Gram-negative uropathogenic isolates, 94.1% (16/17) were found in patients with previous episodes of UTI, who used antibiotics within the previous six months of the starting date of the study, whereas 76.5% (13/17) were obtained from those who had chronic and latent diseases.

## Factors associated with Carbapenem-resistant Gram-negative bacterial uropathogens

In bivariable logistic regression analysis, sex (*P*- 0.107), previous episodes of UTI (*P*- 0.132), history of antibiotic usage in the previous six months (*P*- 0.023), presence of chronic underlying diseases (*P*- 0.001), and hospitalization (*P*- 0.005), had a statistically significant association (Table 5). In addition, multivariable logistic regression analysis showed that for those who used antibiotics during the period of six months prior to the initiation of the study [*P*- 0.048; AOR = 5.045, 95% CI: (1.016–25.053)], the presence of chronic underlying diseases [*P*-0.033; AOR = 3.709, 95% CI: (1.109–12.407)] and a history of hospitalization within the previous one year of the commencement of the study period [*P*-0.031; AOR = 5.760, 95% CI: (1.173–28.276)], were statistically significant (Table 6).

**Table 5. Phenotypically suspected and confirmed carbapenemase producing GN uropathogens from UTI patients.**

| Uropathogens | MDR | Imipenem resistant | Meropenem resistant | Carbapenemase producers |
|---|---|---|---|---|
| | n (%) | n(%) | n(%) | n(%) |
| *E. coli* n = 42 | 21/42(50) | 1/42(2.38) | 3/42(7.14) | 2/42 (4.76) |
| *K. pneumoniae* n = 39 | 22/39(56.4) | 1/39(2.56) | 6/39(15.38) | 5/39 (12.82) |
| *P. aeruginosa* n = 16 | 10/16(62.5) | 2/16(12.5) | 3/16(18.75) | 4/16 (25.0) |
| *P. mirabilis* n = 21 | 10/21(47.6) | 0 | 0 | 0 |
| *Enterobacter* sp. N = 9 | 1/9(11.11) | 0 | 0 | 0 |
| *Citrobacter* sp. N = 4 | 2/4 (50) | 0 | 0 | 0 |
| Total n (%) | 66/131(50.38) | 4/131 (3.1) | 12/131(9.16) | 11/131(8.39) |

**Table 6. Bivariable and multivariable logistic regression analyses of factors associated to carbapenem-resistant Gram-negative uropathogens among patients suspected of UTI attending AMGH, southern Ethiopia, 2020.**

| Variables | Category | Carbapenem resistant | Percentage (%) | P-value | COR [95%CI] | P-value | AOR [95% CI] |
|---|---|---|---|---|---|---|---|
| Sex | Male | 7 | 3.7 | | | | 1 |
| | Female | 10 | 4.3 | **0.107** | 0.402(0.133–1.216) | 0.191 | 0.440(0.129–1.505) |
| Age group | 18–26 | 6 | 5.5 | 0.721 | 1.313(0.295–5.831) | | |
| | 27–35 | 2 | 2.1 | 0.806 | 0.808(0.147–4.423) | | |
| | 36–44 | 4 | 6.3 | 0.271 | 2.545(0.481–13.458) | | |
| | 45–53 | 1 | 1.5 | 0.302 | 0.292(0.28–3.021) | | |
| | >54 | 4 | 4.6 | | 1 | | |
| Residence | Urban | 12 | 4.1 | 0.748 | 1.200(0.394–3.654) | | |
| | Rural | 5 | 3.7 | | 1 | | |
| Educational status | Unable to read and write | 2 | 5.1 | 0.813 | 1.250(0.197–7.942) | | |
| | Primary | 3 | 2.6 | 0.509 | 0.586(0.120–2.861) | | |
| | Secondary | 8 | 4.4 | 0.925 | 1.064(0.292–3.882) | | |
| | College and University | 4 | 4.5 | | 1 | | |
| Marital status | Single | 5 | 4.2 | 0.901 | 0.862(0.082–9.016) | | |
| | Married | 6 | 2.6 | 0.685 | 0.625(0.065–6.046) | | |
| | Divorced | 3 | 6.0 | 0.959 | 0.937(0.079–11.150) | | |
| | Widowed | 3 | 11.5 | | 1 | | |
| Occupation | Government | 2 | 1.6 | | 1 | | |
| | Merchant | 2 | 2.2 | 0.894 | 1.167(0.121–11.254) | | |
| | Farmers | 2 | 5.3 | 0.612 | 0.519(0.041–6.577) | | |
| | Student | 4 | 4.4 | 0.739 | 1.556(0.116–20.854) | | |
| | Home Makers | 5 | 10.6 | 0.870 | 1.217(0.116–12.752) | | |
| | Others | 2 | 7.1 | 0.907 | 1.167(0.089–12.752) | | |
| Monthly income (Birr) | <1000 | 5 | 17.2 | 0.626 | 1.551(0.265–9.071) | | |
| | 1001–2000 | 8 | 14.8 | 0.303 | 0.319(0.036–2.802) | | |
| | >2001 | 4 | 2.6 | | 1 | | |
| Previous exposure to UTI | Yes | 16 | 6.9 | **0.132** | 2.722(0.739–10.026) | 342 | 2.001(0.479–8.358) |
| | No | 1 | 0.5 | | 1 | | 1 |
| History of antibiotic usage in the previous six months | Yes | 16 | 6.1 | **0.023** | 5.859(1.241–26.819) | **0.048**[**] | 5.045(1.016–25.053) |
| | No | 1 | 0.6 | | 1 | | 1 |
| Chronic underlying diseases (diabetes milletus and hypertension) | Yes | 13 | 10.4 | **0.001** | 7.042(2.146–23.105) | **0.033**[**] | 3.709(1.109–12.407) |
| | No | 4 | 1.3 | | 1 | | 1 |
| Hospitalization | Yes | 16 | 10.5 | **0.005** | 6.832(1.887–39.492) | **0.031**[**] | 5.760(1.173–28.276) |
| | No | 1 | 0.4 | | | | 1 |

Note: [*]Statistically significant at $P < 0.25$ in bivariable analysis; [**]Statistically significant at $P < 0.05$; AOR: Adjusted odds ratio; COR: Crude odds ratio; 1: reference group; CI: Confidence interval.

## Discussions

The aetiological agents and their antimicrobial susceptibility profiles may vary over time and among different geographic locations. Our study provides valuable laboratory data on the magnitude, GNB profile, and antimicrobial susceptibility patterns of UTI-suspected cases in AMHG. Besides, we have given the particular emphasis on the prevalence of CRGNB-associated UTI. In this study, the overall prevalence of UTI among the suspected patients was 30.6%

[95% CI: (33.7, 38.3)]. It is slightly higher (27.6%) than that reported in an earlier study conducted in Kenya [28]. Variations in prevalence rates reported in the literature and in this study could also be correlated to the differences in the methodology employed, study population and treatment strategies.

In our study, *E. coli* was the most predominantly isolated organism, followed by *K. pneumoniae*. In several studies, the former (coliuria) was by and large, the most common organism associated with UTI [29–32]. *Proteus mirabilis* and *P. aeruginosa* were other commonly isolated organisms, and this is in line with the results of several studies done in Morocco [29], Uganda [30] and Pakistan [31].

Resistance to commonly used antibiotics is a major concern across the globe, which hampers the successful treatment of UTIs. As antibiotic resistance among uropathogens fluctuates over time and place, regular surveillance and monitoring are crucial in providing healthcare professionals with up-to-date information on the most effective regimen of empirical treatment. Antimicrobial susceptibility test results can also provide information on decreases in the susceptibility of bacteria to various antimicrobials, which can be used to adjust or discontinue the use of the antibiotics for a time, reserving the antibiotics for specific patients, using the drug in combination with another, and generally using an antimicrobial from a different class to delay or avoid the intensity of drug resistance. In our study, the antibiogram of GNB isolates showed varied profiles. The highest resistance was shown against ampicillin, followed by piperacillin and tobramycin. This trend is pretty similar to that found in some earlier studies conducted in Morocco [29], Egypt [33], and another city in Ethiopia [34]. Presently, amoxicillin and ampicillin are not recommended as reliable agents for empirical therapy because higher levels of resistance to these drugs have been reported for over two decades, and also it is well reflected in this study and a couple of our previous works done in the title hospital [35, 36].

A moderate degree of resistance was observed against the first-line drug, ciprofloxacin (46.6%), used to manage uncomplicated UTI in the title hospital. This finding hints at a rising resistance against ciprofloxacin. At the same time, only a marginally lower degree of resistance was detected against the first-line alternative drugs such as trimethoprim-sulfamethoxazole and nitrofurantoin. These medications appear to be an option for the treatment of uncomplicated UTI in the study area. It is interesting to note that only 3.8 and 9.2% of isolates were resistant to imipenem and meropenem, respectively. These results resemble the outcome of some of the earlier studies conducted in Pakistan [31] and Libya [37], but they are not in line with another study done in Iraq, which showed that the resistance to carbapenem was to the extent of 23.1% [38]. Such variations could be attributed to differences in the types of isolated organisms and specimens collected.

With regard to the predominant uropathogenic *E. coli*, invariably, all the isolates displayed resistance against ampicillin, which was at par with the results already documented in a previous study done in another city in Ethiopia [34]. At the same time, they exhibited a moderate degree of resistance to three antibiotics such as ciprofloxacin (38%), trimethoprim-sulfamethoxazole (33.3%) and nitrofurantoin (26.2%). This result resembles that of a couple of previous studies done in Morocco [29] and Egypt [33]. Interestingly, these isolates were extremely susceptible (>90%) to both carbapenem drugs tested, and this is similar to the results of some studies done in Libya [37], and Pakistan [39].

The second most predominant bacteria, *K. pneumoniae*, showed varying degrees of resistance to different antibiotics, i.e., piperacillin, 84.6%, tobramycin, 79.5% and ciprofloxacin, 59%. The higher degree of resistance observed against these drugs is concerning, and therefore, the antibiotic prescription policies must be reviewed in the study area. A similar pattern of resistance profile was observed in a previous study conducted in Italy [40]. More than 80% of the isolates of *P. aeruginosa* were resistant to a couple of antibiotics, such as piperacillin and

tobramycin, and this is similar to an earlier trend observed in Pakistan [41]. Our current findings strongly indicate that empirical treatment with these antibiotics should be discouraged.

Multidrug resistance among uropathogens is a major factor undermining the possibilities of empirical therapy. In our study, MDR was detected in the case of 50.38% of isolates, which is slightly higher than that found in a study already done in Ethiopia (46.2%) [42] and at the same time, lower than that reported in another work conducted in the northern part of the country [87.4%] [19]. Gradation in the extent of resistance may be due to the difference in sample size, study design, prescription patterns, and antibiotic therapy practised. In addition, the probability of uropathogenic bacterial isolates evolving into MDR strains over time depends on several factors, including biofilm formation in the bladder, the emergence of ESBL-producing strains, the misuse and overuse of antibiotics by unqualified practitioners, and the easy accessibility of antibiotics [43]. Stringent antimicrobial stewardship policies and the prudent use of antimicrobials can help to reduce the spread of antimicrobial-resistant genes in the hospital setting.

The overall prevalence of CRGNB uropathogens in this study was 12.98% [95% CI: (7.0, 19.0)]. The higher isolation rate of CRGNB is an alert sign for clinicians and medical microbiologists to implement strict surveillance and also more stringent infection control measures in hospital settings and in general. This alarming trend may be due to the over usage and inappropriate dosage of carbapenem, which has triggered organisms to develop resistance mechanisms over time. The prevalence of CRGNB uropathogens observed in this study is in jibe with those found in a series of related works done earlier in different countries, including Ethiopia (12.12%), Greece (7.5%), Italy (7.4%), Russia (11.0%) and Nigeria (12.4%) [19, 44, 45]. A possible explanation for this parity could be correlated to the similarity in the types of bacteria isolated (Enterobacteriaceae) irrespective of the place of work.

On the other hand, our findings show an upward trend compared to the results of studies conducted in several other countries; for instance, the prevalence of CRGNB uropathogens found by us is more than that existing in Romania (5.0%) [45], USA (5.7%) [46], and Morocco (6.4%) [27]. This discrepancy could be attributed to the absence of standard infection control and prevention measures, the high incidence of carbapenem-resistant uropathogens in developing countries, and also to the wide variations in socio-demographic characteristics. The current set of results obtained is not in tandem with the outcome of a study on prevalence conducted in Uganda (28.6%) [30]. A possible explanation for this inconsistency could match with the differences in factors associated with the study participants, such as socio-economic circumstances, conditions favouring CRGNB, and the extent of consumption of carbapenem drugs, especially in several nations with a high prevalence of CRE, and having some geographical peculiarities.

In the Enterobacteriaceae family, carbapenemase production has been found most commonly in *E. coli* and *K. pneumoniae* [46]. In our study, *K. pneumoniae* were the predominant isolates among CRGNB, of which the majority were carbapenemase producers. According to the European Centre for Disease Prevention and Control, up to 62% of *Klebsiella* spp. are resistant to carbapenems [47]. As per the WHO Global Report on antimicrobial resistance surveillance, carbapenem resistance shown by Klebsiella species (more than 50%) was reported from two regions belonging to the group of 71 WHO member states [48].

The second most dominant isolates of CRGNB were *P. aeruginosa*, of which a more significant proportion was carbapenemase producers. These results are analogous to the outcome of several studies conducted in China [49] and various African countries [50]. In this study, *E. coli* is the least isolated CRGNB, of which only fifty percent were producers of carbapenemase. This result contradicts a study done in Morocco, where *E. coli* was the second most prominently isolated CRE [29]. The present study also showed that CRGNB and all carbapenemase-

producing isolates are highly resistant to almost all antibiotics tested. This trend was similar to a study conducted earlier in Ethiopia which showed that all the isolates of CRE were resistant to trimethoprim-sulfamethoxazole (100%) and ciprofloxacin (85–100%) [19]. Furthermore, our findings indicated that the resistance profiles exhibited by isolates of *E. coli*, *P. mirabilis*, *K. pneumoniae*, and *P. aeruginosa* were more or less in parity with the results of previous studies carried out in a couple of locations in Ethiopia, such as Hawassa [34] and Addis Ababa [51]. The increasing resistance exhibited by these pathogens pave the way for other co-infections and treatment delay. The alarming rates of resistance of uropathogens to commonly used antibiotics necessitate a re-evaluation of empirical treatment guidelines, highlighting the need to prevent antibiotic abuse in daily practice in the title hospital.

Different socio-demographic and clinically associated factors were analyzed by considering their impact on the spread of carbapenem resistance among Gram-negative uropathogens. Of those factors, the usage of antibiotics during the previous six months of the commencement of the study was one of the most significantly associated factors. Patients who had a history of usage of antibiotics were five times more prone to be infected by CRGNB uropathogens; this is similar to some studies conducted in China [49], Egypt [33], and even in another city of Ethiopia [19]. The likely reasons for this similarity could be correlated to the trend of antibiotic usage in early childhood, indiscriminate prescription of antibiotics for treating UTI, higher prevalence of CRE, low adherence to drug regimens, and dependence on empirical therapy. This study showed the strong emergence of carbapenem-resistant GNB, resulting in UTIs. Furthermore, these bacterial uropathogens are resistant to multiple antibiotics tested, which is also a direful situation challenging the practice of empirical treatment.

Another vital factor that had an influential association with the prevalence of CRGNB uropathogens was the incidence of hospitalizations. This scenario can also be correlated to the severity of existing diseases and frequency of exposure. Participants who were hospitalized prior to the initiation of this study were 5.7 times more prone to develop infections associated with CRGNB uropathogens. This aspect is consistent with earlier studies conducted in different nations, including Singapore [52], Egypt [33], and even another city in Ethiopia [19]. Possible reasons include inappropriate regimens prescribed, non-uniform regimens, the presence of autoimmunity, over or under-dosages of antibiotics, and the depletion of protective commensal microbiota.

The third associated factor which can be considered being significant is the presence of chronic underlying diseases. It showed a statistically significant association with the dependent variables included in our study. Participants having chronic underlying diseases (diabetes mellitus and hypertension) had 3.7 fold enhanced chance of developing infections associated with CRGNB uropathogens. This was also shown in a study conducted in Israel which evaluated the risk factors associated with CRE acquisition [53].

Shortcomings of the present study include shorter duration and the type of design of the study (cross-sectional), and a smaller sample size. Besides, only conventional methods of urine culture and carbapenemase enzyme production were employed. Molecular detection of virulence and antimicrobial-resistant genes of the isolates was not performed due to the lack of infrastructure/ facilities.

## Conclusions

This study is the first of its kind concerning CRGNB in Arba Minch, Ethiopia. The prevalence of multidrug-resistant and carbapenem-resistant Gram-negative uropathogens resembled the previous study reported from Ethiopia. It is to be stressed that three scores of carbapenem-resistant isolates were carbapenemase producers. Isolates of *K. pneumoniae* were the

predominant carbapenemase producer. Hospitalization and the history of usage of antibiotics during the previous twelve and six months of the initiation of the study, respectively. The presence of chronic underlying diseases was statistically significant with regard to carbapenem resistance. The majority of isolates were susceptible to carbapenem drugs and nitrofurantoin, whereas ampicillin was found to be the most ineffective drug as per the present set of data. Therefore, providing adequate awareness programs on antimicrobial resistance, clubbed with stringent antimicrobial stewardship policies and prudent use of antimicrobials by clinicians, are of utmost importance.

## Supporting information

**S1 Dataset.**
(XLS)

## Acknowledgments

The authors thankfully acknowledge the Department of Medical Laboratory Science, College of Medicine and Health Sciences, Arba Minch University, for the facilities. Thanks are extended to Prof. Dr. K. R. Sabu for English-language editorial work.

## Author Contributions

**Conceptualization:** Asaye Mitiku, Addis Aklilu, Tsegaye Tsalla, Melat Woldemariam.

**Data curation:** Asaye Mitiku, Addis Aklilu, Tsegaye Tsalla, Melat Woldemariam.

**Formal analysis:** Asaye Mitiku, Addis Aklilu, Tsegaye Tsalla, Melat Woldemariam.

**Investigation:** Asaye Mitiku, Addis Aklilu, Tsegaye Tsalla, Melat Woldemariam.

**Methodology:** Asaye Mitiku, Addis Aklilu, Tsegaye Tsalla, Melat Woldemariam, Aseer Manilal, Melkam Biru.

**Project administration:** Asaye Mitiku, Addis Aklilu, Tsegaye Tsalla, Melat Woldemariam.

**Resources:** Asaye Mitiku, Addis Aklilu, Tsegaye Tsalla, Melat Woldemariam, Melkam Biru.

**Software:** Asaye Mitiku, Addis Aklilu, Tsegaye Tsalla, Melat Woldemariam, Aseer Manilal, Melkam Biru.

**Supervision:** Addis Aklilu, Tsegaye Tsalla, Melat Woldemariam.

**Validation:** Asaye Mitiku, Addis Aklilu, Tsegaye Tsalla, Melat Woldemariam, Melkam Biru.

**Visualization:** Asaye Mitiku, Addis Aklilu, Tsegaye Tsalla, Melat Woldemariam, Aseer Manilal, Melkam Biru.

**Writing – original draft:** Asaye Mitiku, Addis Aklilu, Melat Woldemariam, Aseer Manilal, Melkam Biru.

**Writing – review & editing:** Addis Aklilu, Melat Woldemariam, Aseer Manilal.

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
