## [Decision Letter · Decision Letter 0]

4 Oct 2022

PONE-D-22-08147Carbapenemase Producing Gram Negative bacterial pathogens and Associated Factors among Patients Suspected of Urinary Tract Infections Attending Arba Minch General Hospital, southern EthiopiaPLOS ONE

Dear Dr. Manilal,

Thank you for submitting your manuscript to PLOS ONE. After careful consideration, we feel that it has merit but does not fully meet PLOS ONE’s publication criteria as it currently stands. Therefore, we invite you to submit a revised version of the manuscript that addresses the points raised during the review process.

We look forward to receiving your revised manuscript.

Kind regards,

Guadalupe Virginia Nevárez-Moorillón, Ph.D.

Academic Editor

PLOS ONE

Journal Requirements:

Additional Editor Comments (if provided):

Please, consider all the comments by the reviewers, and revise the document for language structure and grammar.

Reviewers' comments:

Reviewer's Responses to Questions

**Comments to the Author**

1. Is the manuscript technically sound, and do the data support the conclusions?

Reviewer #1: Yes

Reviewer #2: Partly

Reviewer #3: No

2. Has the statistical analysis been performed appropriately and rigorously? 

Reviewer #1: I Don't Know

Reviewer #2: Yes

Reviewer #3: Yes

3. Have the authors made all data underlying the findings in their manuscript fully available?

Reviewer #1: No

Reviewer #2: Yes

Reviewer #3: Yes

4. Is the manuscript presented in an intelligible fashion and written in standard English?

Reviewer #1: No

Reviewer #2: No

Reviewer #3: No

5. Review Comments to the Author

Reviewer #1: The manuscript “Carbapenemase Producing Gram Negative bacterial pathogens and Associated Factors among Patients Suspected of Urinary Tract Infections Attending Arba Minch General Hospital, southern Ethiopia” by Manilal et al contains valuable information about antimicrobial resistance in gram negative uropathogens isolated from clinical patients.

Comments

• Title focused on CP-CRE although the content of the manuscript is not specific to CP-CRE and they assessed susceptibility to other antimicrobials too?

Abstract: How did you test for carbapnemase production? Line 44 Delete "Isolates of " “Consumption of antibiotics prior to six months of the commencement of the study, presence of chronic diseases and hospitalizations were statistically significant" with what ??

• Introduction: Line 65-67, not clear please re-write.

• What was the basis for suspecting a patient to have UTI? Please elaborate this in methods section? Are these patients Inpatient or outpatient?

• standard calibrated wire loop? Do you calibrate wire loop? how?

• Line 139: cotrimoxazole?? Do you mean Sulphametoxazole + trimetoprim??

• Results Page 10: Line 184-186"14% (17) were resistant to both imipenem and meropenem. Of them, 29.4 % (5/17) were found to be imipenem resistant whereas the remaining 70.6 % (12/17) were meropenem resistant" this is not clear. ??? If 17 are resistant to both then how comes only 12 to one?

• Line 200: previous exposure to UTI??? Do you mean previous history of UTI?

• Heading of Table 4 should be re-written. It should not be resistance patter rather antimicrobial susceptibility profile

• I doot know why the authors wanted to test susceptibility of Klebsiella species for their susceptibility to ampiicillin, this species is intrinsically resistant to ampicillin. What is the basis of selection of the nine antimicrobials?

• Please list the chronic underlying diseases described as risk factors ? which are these diseases? Elaborate these in the methods section including how you collect these information?

• In addition it is not clear how you managed to know if the patients had previous history of antimicrobial use.

• Is it just by interviewing the patient? Do they all differentiate if they have taken antibiotics or other medication

Reviewer #2: The manuscript entitled "Carbapenemase producing gram negative bacterial pathogens and associated factor among patioents suspected of urinary tract infections attending Arba Minch General Hospital, southern Ethiopia" report the frecuency of carban¡penem resistant bacteria in patiens with urinary infection. The results are only descriptive and I can't found any significant contribution to the clinical microbiologycal field. Althogh, the authors mentioned that this is one of the first resport of carbapenem resistant bacteria in Ethiopia, there are some other works which reports the prevalence of this microorganisms in the same zone (Alemayehu et al., 2021; Tekele et al., 2021; Desta et al., 2016; Legese et al., 2017... among others). For this reason I can not recommended their publication in PLOS ONE

Reviewer #3: Title: Carbapenemase Producing Gram Negative bacterial pathogens and Associated Factors among Patients Suspected of Urinary Tract Infections Attending Arba Minch General Hospital, southern Ethiopia

I would like to appreciate the Editors of PLOS One invitation to contribute to reviewing process of this interesting study.

Comments

This article This study determined the prevalence of carbapenemase producing Gram-negative uropathogens and associated factors among patients suspected of urinary tract infections. however, there are multiple issues with this manuscript.

Introduction of the study: Failed to provide adequate background for the study. the rationale of the study and the research problem not well stated. There are incomplete and lack of logical flow of the sentences and paragraphs. Also, the scientific literature not well cited (e. g from page 78 to 86).

Methodology: The authors used a routine investigation of carbapenemase production. It would be better to perform molecular characterization of carbapenemase genes of blaOXA-48 like, bla NDM-1 and other common genes, which are now become predominant determinants to cause carbapenem resistant in Gram-negative pathogens worldwide.

Results, Discussions and Abstract: to be modified accordingly.

Others

The writing style needs to be modified.

6. PLOS authors have the option to publish the peer review history of their article (what does this mean?). If published, this will include your full peer review and any attached files.

Reviewer #1: No

Reviewer #2: No

Reviewer #3: No

---

## [Author Response · Author response to Decision Letter 0]

3 Dec 2022

Reviewer 1 

The manuscript “Carbapenemase Producing Gram Negative bacterial pathogens and Associated Factors among Patients Suspected of Urinary Tract Infections Attending Arba Minch General Hospital, southern Ethiopia” by Manilal et al contains valuable information about antimicrobial resistance in gram negative uropathogens isolated from clinical patients.

 Thanks for your encouraging comments. 

We have revised the manuscript substantially as per your comments and suggestions

Title focused on CP-CRE although the content of the manuscript is not specific to CP-CRE and they assessed susceptibility to other antimicrobials too? The title has been modified as ‘Magnitude and antimicrobial susceptibility profiles of Gram-Negative bacterial isolates among Patients Suspected of Urinary Tract Infections in Arba Minch General Hospital, southern Ethiopia.’

Abstract: 

 How did you test for carbapnemase production? 

Line 44 Delete "Isolates of " 

“Consumption of antibiotics prior to six months of the commencement of the study, presence of chronic diseases and hospitalizations were statistically significant" with what ?? Thanks for your valuable comment. 

Potential carbapenemase producers were phenotypically determined by the modified carbapenem inactivation method as per the CLSI guidelines.

The abstract part is substantially revised and includes a line on carbapenemase production as per your suggestion. 

Deteled in the revised manuscript

Consumption of antibiotics prior to six months of the commencement of the study, presence of chronic diseases and hospitalizations were statistically significant with UTI caused by CP-CRE.

INTRODUCTION 

Introduction: Line 65-67, not clear please re-write. Thanks for your good support. 

The lines are modified in the revised manuscript.

MATERIALS AND METHODS 

What was the basis for suspecting a patient to have UTI? Please elaborate this in methods section? 

Are these patients Inpatient or outpatient?

standard calibrated wire loop? Do you calibrate wire loop? how?

Line 139: cotrimoxazole?? Do you mean Sulphametoxazole + trimetoprim??

Corrected in the revised manuscript as

inclusion criterion is all adult patients clinically suspected of UTI (ie., frequency, urgency, dysuria and hematuria) 

All adult patients suspected of UTI who attended the Outpatient Department. 

Corrected in the revised manuscript.

We have used 1 µL disposable calibrated plastic loop.

Yes, it is trimethoprim-sulfamethoxazole (1.25/23.75 µg)

We have replaced it as and where required in the revised manuscript.

RESULTS

Results Page 10: Line 184-186"14% (17) were resistant to both imipenem and meropenem. Of them, 29.4 % (5/17) were found to be imipenem resistant whereas the remaining 70.6 % (12/17) were meropenem resistant" this is not clear. ??? If 17 are resistant to both then how comes only 12 to one?

Line 200: previous exposure to UTI??? Do you mean previous history of UTI?

Heading of Table 4 should be re-written. It should not be resistance patter rather antimicrobial susceptibility profile

I doot know why the authors wanted to test susceptibility of Klebsiella species for their susceptibility to ampiicillin, this species is intrinsically resistant to ampicillin. 

What is the basis of selection of the nine antimicrobials?

Please list the chronic underlying diseases described as risk factors ? which are these diseases? Elaborate these in the methods section including how you collect these information?

• In addition it is not clear how you managed to know if the patients had previous history of antimicrobial use.

• Is it just by interviewing the patient? 

Do they all differentiate if they have taken antibiotics or other medication

Reviewer 2

Reviewer #2: The manuscript entitled "Carbapenemase producing gram negative bacterial pathogens and associated factor among patioents suspected of urinary tract infections attending Arba Minch General Hospital, southern Ethiopia" report the frecuency of carban¡penem resistant bacteria in patiens with urinary infection. The results are only descriptive and I can't found any significant contribution to the clinical microbiologycal field. Althogh, the authors mentioned that this is one of the first resport of carbapenem resistant bacteria in Ethiopia, there are some other works which reports the prevalence of this microorganisms in the same zone (Alemayehu et al., 2021; Tekele et al., 2021; Desta et al., 2016; Legese et al., 2017... among others). For this reason I can not recommended their publication in PLOS ONE

Reviewers 3

Reviewer #3: Title: Carbapenemase Producing Gram Negative bacterial pathogens and Associated Factors among Patients Suspected of Urinary Tract Infections Attending Arba Minch General Hospital, southern Ethiopia

I would like to appreciate the Editors of PLOS One invitation to contribute to reviewing process of this interesting study.

Comments

This article This study determined the prevalence of carbapenemase producing Gram-negative uropathogens and associated factors among patients suspected of urinary tract infections. however, there are multiple issues with this manuscript.

Introduction of the study: Failed to provide adequate background for the study. the rationale of the study and the research problem not well stated. There are incomplete and lack of logical flow of the sentences and paragraphs. Also, the scientific literature not well cited (e. g from page 78 to 86).

Methodology: The authors used a routine investigation of carbapenemase production. It would be better to perform molecular characterization of carbapenemase genes of blaOXA-48 like, bla NDM-1 and other common genes, which are now become predominant determinants to cause carbapenem resistant in Gram-negative pathogens worldwide.

Results, Discussions and Abstract: to be modified accordingly.

 Thanks for your useful comment

The typo error has been corrected in the revised manuscript

Yes, it is a previous episode of UTI.

The typo error has been corrected. 

Corrected in the revised manuscript

Thanks for your invaluable comment

Corrected in the revised manuscript

These were selected based on CLSI, considering the national antibiotic policy and also based on the availability. 

Thanks for your invaluable comment.

We have collected the clinical data after reviewing the patients' records.

Chronic diseases are diabetes mellitus, hypertension and renal diseases. In the final analysis, 13 cases comprised of DM, and hypertension, along with the appearance of both cases, are observed in a few patients. Therefore we have merged it as a single variable.

We have extracted this information from reviewing the patients' record.

No, it is data extracted from the patient record.

Yes, the patients were differentiated and excluded as per the inclusion criteria set ie.,

(patients who had received antibiotics within the previous two weeks of the commencement of the study)

Thanks for your comments. 

CRE is a highly prioritized pathogens enlisted by the WHO. Therefore, studies related this topic is quite pertinent.

Thanks for your supportive comment

We have extensively modified the background and discussion part.

Thanks for your valid points

However, due to the lack of infrastructure and facility, we are unable to perform molecular detection of genes. This is included under the limitation as per your suggestions.

Results and discussion is a revised manuscript

---

## [Editor Report · Decision Letter 1]

19 Dec 2022

Magnitude and antimicrobial susceptibility profiles of Gram-Negative bacterial isolates among Patients Suspected of Urinary Tract Infections in Arba Minch General Hospital, southern Ethiopia

PONE-D-22-08147R1

Dear Dr. Manilal,

We’re pleased to inform you that your manuscript has been judged scientifically suitable for publication and will be formally accepted for publication once it meets all outstanding technical requirements.

Kind regards,

Guadalupe Virginia Nevárez-Moorillón, Ph.D.

Academic Editor

PLOS ONE
---

## [Editor Report · Acceptance letter]

21 Dec 2022

PONE-D-22-08147R1 

Magnitude and antimicrobial susceptibility profiles of Gram-Negative bacterial isolates among Patients Suspected of Urinary Tract Infections in Arba Minch General Hospital, southern Ethiopia 

Dear Dr. Manilal:

I'm pleased to inform you that your manuscript has been deemed suitable for publication in PLOS ONE. Congratulations! Your manuscript is now with our production department. 

Kind regards, 

on behalf of

Dr. Guadalupe Virginia Nevárez-Moorillón 

Academic Editor

PLOS ONE